# Benefits from Adjuvant Chemotherapy in Patients with Resected Non-Small Cell Lung Cancer: Possibility of Stratification by Gene Amplification of *ACTN4* According to Evaluation of Metastatic Ability

**DOI:** 10.3390/cancers14184363

**Published:** 2022-09-07

**Authors:** Takehiro Tozuka, Rintaro Noro, Masahiro Seike, Kazufumi Honda

**Affiliations:** 1Department of Pulmonary Medicine and Oncology, Graduate School of Medicine, Nippon Medical School, Tokyo 113-8603, Japan; 2Department of Bioregulation, Graduate School of Medicine, Nippon Medical School, Tokyo 113-8602, Japan; 3Institution for Advanced Medical Science, Nippon Medical School, Tokyo 113-8602, Japan

**Keywords:** non-small cell lung cancer, adjuvant chemotherapy, biomarker, actinin-4, metastatic ability

## Abstract

**Simple Summary:**

The establishment of biomarkers that can identify individuals at high risk of early recurrence after surgery will be an important issue in decision-making for perioperative therapy. In this review, we describe potential biomarkers for predicting the likelihood of recurrence in patients who undergo surgery for stage I NSCLC. ACTN4 is a possible biomarker for identifying patients at high risk of postoperative recurrence, and patients with gene amplification of *ACTN4* might thus benefit the most from adjuvant chemotherapy.

**Abstract:**

Surgical treatment is the best curative treatment option for patients with non-small cell lung cancer (NSCLC), but some patients have recurrence beyond the surgical margin even after receiving curative surgery. Therefore, therapies with anti-cancer agents also play an important role perioperatively. In this paper, we review the current status of adjuvant chemotherapy in NSCLC and describe promising perioperative therapies, including molecularly targeted therapies and immune checkpoint inhibitors. Previously reported biomarkers of adjuvant chemotherapy for NSCLC are discussed along with their limitations. Adjuvant chemotherapy after resective surgery was most effective in patients with metastatic lesions located just outside the surgical margin; in addition, these metastatic lesions were the most sensitive to adjuvant chemotherapy. Thus, the first step in predicting patients who have sensitivity to adjuvant therapies is to perform a qualified evaluation of metastatic ability using markers such as actinin-4 (ACTN4). In this review, we discuss the potential use of biomarkers in patient stratification for effective adjuvant chemotherapy and, in particular, the use of ACTN4 as a possible biomarker for NSCLC.

## 1. Introduction

Lung cancer is the leading cause of cancer death worldwide [1]. (NSCLC) non-small cell lung cancer (NSCLC) accounts for 85% of all lung cancers [2]. Almost 20% of patients have early-stage disease (stage I–II) and around 30% of patients have locally advanced disease (stage III) at the time of diagnosis of NSCLC [3]. Surgical resection is the best curative treatment option for patients with NSCLC. However, local or systemic relapse of the disease is common despite complete resection. Five-year overall survival (OS) rates are reported to be about 70%, 50–60%, and 35% for pathological stages IB, IIA–IIB, and IIIA NSCLC, respectively [4].

Adjuvant chemotherapy is performed to prevent recurrence in patients who undergo complete resection of NSCLC. Adjuvant cisplatin-based chemotherapy is the standard treatment for postoperative patients with stage IIA–IIIA NSCLC. Pooled analysis of several clinical trials has shown that adjuvant cisplatin-doublet chemotherapy improved the 5-year disease-free survival (DFS) rate by 5.8% and the 5-year OS rate by 5.4% [5] Moreover, the IMpower 010 trial demonstrated that atezolizumab after adjuvant platinum-based chemotherapy improved DFS in postoperative patients who were selected by programmed death-ligand 1 [PD-L1] expression [6]. The ADAURA trial demonstrated that osimertinib improved DFS in epidermal growth factor receptor (EGFR) mutation-positive stage IB–IIIA disease following complete resection [7]. However, a pooled analysis of several clinical trials showed that some patients without adjuvant chemotherapy achieved five-year DFS and that the rate of overall grade 3 to 4 toxicity was 66% in patients with adjuvant platinum-based chemotherapy [5]. As a result, some patients with lung cancer were cured by chest surgery alone, and adjuvant chemotherapy may be overtreatment for such patients [5]. An economic analysis of a randomized trial of adjuvant vinorelbine plus cisplatin suggested a high cost-effectiveness of this combined treatment compared with other standard healthcare interventions [8]. However, a pilot study revealed that patients undergoing cancer treatment may change their treatment to defray out-of-pocket costs because of financial burden [9]. In particular, molecularly targeted therapies and immune checkpoint inhibitors (ICIs) may increase economic toxicity for cancer patients due to the high prices of these drugs [10]. From the perspective of reducing adverse events and saving on the medical costs of adjuvant chemotherapy, biomarkers that can predict the likelihood of recurrence after surgery would be of great interest. In contrast, some patients with completely resected NSCLC may have minimal residual disease (MRD) that is undetectable radiographically due to the limits of imaging resolution. Therefore, it is important to establish biomarkers that can predict the risk of recurrence of lung cancer. To date, many studies have investigated specific gene expression, gene signature, and excision repair cross-complementation group 1 (ERCC1), which is related to cisplatin (CDDP) sensitivity, as candidate biomarkers for adjuvant chemotherapy in patients with NSCLC [11,12,13,14,15,16,17,18,19,20,21,22,23,24,25]. However, a robust biomarker has not yet been established and clinical application has been challenging because it is difficult to quantify the biomarkers and determine the cutoff values. It is generally considered that patients with MRD are the best candidates for adjuvant chemotherapy, as the presence of MRD beyond the surgical margin is known to be a strong indicator of the metastatic ability of the primary site. Thus, there is a need for biomarkers that can accurately evaluate the metastatic ability of the primary site and thus help decide the strategy for precision adjuvant chemotherapy.

Actinin-4 (ACTN4) is an anti-binding protein involved in cancer invasion and metastasis [23]. In various cancers, including lung cancer, increased protein expression of ACTN4 indicates malignancy and metastatic potential [26,27,28]. Gene amplification of *ACTN4* located on 19q13 is also significantly associated with metastatic potential in various types of cancer [29,30,31]. We have recently reported that increased protein expression and gene amplification of ACTN4 can be a promising biomarker for adjuvant chemotherapy in postoperative patients with NSCLC [32,33,34]. In this review, we discuss the role of ACTN4 in the process of tumor progression and the usefulness of ACTN4 as a potential biomarker for selecting patients most likely to benefit from adjuvant chemotherapy.

## 2. Adjuvant Chemotherapy in NSCLC

Previous clinical trials have demonstrated improvements in DFS and OS following adjuvant cisplatin-based chemotherapy for resected NSCLC [35,36,37]. A meta-analysis including five clinical trials also showed that cisplatin-based adjuvant chemotherapy prolonged OS compared with surgery alone (hazard ratio [HR], 0.89; 95% confidence interval [CI], 0.82–0.96). Adjuvant cisplatin-doublet chemotherapy achieved a 5.4% improvement in the 5-year OS rate [5]. However, some of these patients had severe adverse events (AEs). In the meta-analysis, the rates of grade 3 or 4 AEs and of grade 4 AEs were reported to be 66% and 32%, respectively. Although toxicity types differed according to the chemotherapeutic regimen, the most frequent AE was neutropenia. Treatment-related deaths were reported as 0.9%. Subset analysis by stage of cancer showed possible harm in stage IA (HR, 1.40; 95%CI, 0.95–2.06) [5]. The clinical benefits of adjuvant chemotherapy tend to be higher at more advanced stages [5]. In Japan, adjuvant therapy with tegafur/uracil (UFT) is recommended for patients with stage IA–IB NSCLC after complete resection [38]. A meta-analysis of adjuvant UFT in NSCLC demonstrated that surgery plus UFT prolonged OS compared with surgery alone (HR, 0.74; 95%CI, 0.61–0.88) [38,39].

Molecular targeted therapy has been a standard treatment for advanced NSCLC patients with driver mutations such as EGFR mutation or anaplastic lymphoma kinase (ALK) rearrangement. Clinical trials of adjuvant chemotherapy using tyrosine kinase inhibitors (TKIs) have been conducted perioperatively in patients with EGFR-mutant NSCLC. In the RADIANT randomized controlled trial, Erlotinib was compared to placebo in patients with completely resected stage IB-IIIA NSCLC [40]. In a subgroup analysis of the RADIANT trial, erlotinib prolonged DFS (HR, 0.61; 95%CI: 0.38–0.98) but not OS (HR, 1.09; 95%CI: 0.55–2.16) compared to placebo in patients with EGFR-mutant NSCLC. The ADJUVANT trial compared gefitinib (oral for two years) with cisplatin plus vinorelbine (four cycles) in stage II–IIIA NSCLC patients with common EGFR mutations (exon 19 deletion or exon 21 L858R mutation) after complete resection [41]. DFS was significantly longer in the gefitinib group (HR, 0.60; 95%CI, 0.42–0.87), but there was no significant difference in OS between the groups (HR, 0.92; 95%CI, 0.62–1.36). In the IMPACT trial, adjuvant gefitinib (oral for two years) did not prolong DFS and OS compared with cisplatin plus vinorelbine (four cycles) in patients with completely resected Stage II–IIIA NACLC with common EGFR mutations [42]. The ADAURA trial compared osimertinib (oral for three years) with placebo in stage IB–IIIA NSCLC patients with common EGFR mutations after complete resection. In that study, the physician decided whether patients received adjuvant platinum-based chemotherapy before administration of osimertinib or placebo. Adjuvant osimertinib significantly prolonged DFS compared with placebo (HR, 0.17; 99.06%CI, 0.11–0.26), although OS was immature [7]. These results suggest that adjuvant chemotherapy using EGFR-TKI may become a treatment option in the near future, but toxicity and medical cost should be considered. Therefore, it is necessary to identify biomarkers for detecting patients with a high risk of recurrence and those who would benefit from adjuvant EGFR-TKI.

ICIs including anti-programmed cell death 1 (PD-1) antibodies, anti-PD-L1 antibodies, and anti-cytotoxic T-lymphocyte-associated protein 4 (CTLA-4) antibodies have greatly improved prognosis in patients with metastatic NSCLC. ICIs have recently moved from the second-line to the first-line setting for metastatic NSCLC patients without driver mutations [43,44,45,46,47]. At present, ICIs play an important role in the treatment of locally advanced NSCLC. The PACIFIC trial revealed that durvalumab after chemoradiotherapy prolonged progression-free survival (PFS) and OS in unresected stage IIIA NSCLC patients [48]. Moreover, ICI has become a promising treatment in perioperative patients with NSCLC. The IMpower 010 trial demonstrated that atezolizumab after adjuvant platinum-based chemotherapy significantly prolonged DFS in postoperative patients with PD-L1 (SP263) positive stage II–IIIA NSCLC (HR, 0.66; 95%CI, 0.50–0.88) [6], but there were no survival benefits in patients without PD-L1 expression (HR, 0.97; 95%CI, 0.72–1.31). Neoadjuvant chemotherapy has not become a standard perioperative treatment in patients with NSCLC because it can lead to increased perioperative complications [49]. However, there are many clinical trials of neoadjuvant chemotherapy using ICIs, and neoadjuvant immunotherapy is a promising treatment [50]. The Checkmate 816 trial showed that neoadjuvant nivolumab plus platinum-based chemotherapy achieved a significantly longer event-free survival (HR, 0.63; 97.38%CI, 0.43–0.91; *p* = 0.005) and a higher pathological complete response rate (odds ratio, 13.94; 99%CI, 3.49–55.75) compared with neoadjuvant platinum-based chemotherapy alone in patients with stage IB-IIIA NSCLC [51]. However, neoadjuvant chemotherapy in patients with NSCLC can induce severe adverse events preoperatively that may cause surgery to be postponed or canceled. In the Checkmate 816 trial, definitive surgery was canceled in 16% of patients included in the nivolumab plus chemotherapy arm. The reasons for canceled surgery were disease progression (7%), AEs (1%), or other scenarios (8%). Patient selection by appropriate biomarkers is important in both the adjuvant and neoadjuvant settings.

## 3. Current Biomarker Candidates for Perioperative Patients with NSCLC

Many previous studies have attempted to identify useful biomarkers for adjuvant chemotherapy, but none have yet been established in clinical practice.

Numerous studies have reported that signatures based on gene expression are prognostic factors in patients with NSCLC [11,12,13,14,15,16,17,18,19,20,21,22]. However, none of these signatures are ready for clinical application as they require statistical validation and reproducibility of the signatures. In addition, their actual medical utility and medical cost are unknown. It is also difficult to standardize the quantification methods and set cutoff levels because most studies have been based on microarray analysis of mRNA expression levels [11,12,13,14,15,16,17,18,19,20,21,22]. There is no consensus regarding the usefulness of genetic analysis in determining a treatment strategy for patients with postoperative lung cancer.

High expression of ERCC1 was reported to be a prognostic factor in patients with early-stage NSCLC who had received surgery alone [23]. DNA repair capacity is strongly associated with cisplatin resistance. In particular, the ERCC1 protein is considered to play an important role in nucleotide excision repair [24]. Therefore, several studies have investigated the association between ERCC1 expression and the efficacy of adjuvant platinum-based chemotherapy. A previous study suggested that ERCC1 was related to the resistance of NSCLC to cisplatin-based chemotherapy [24]. It included patients with completely resected NSCLC and found that patients with ERCC1-negative tumors benefited from adjuvant cisplatin-based chemotherapy but patients with ERCC1-positive tumors did not [24]. However, another study that performed immunohistochemical analysis of patients in two independent phase 3 trials was unable to validate ERCC1 protein expression as a predictive marker for the efficacy of adjuvant-chemotherapy [25]. Therefore, the usefulness of ERCC1 expression has not been established in therapeutic decision-making for patients with completely resected NSCLC.

Previous studies have verified the prognostic and predictive effects of the tumor suppressor gene tumor protein p53 gene (TP53) in patients with completely resected NSCLC. TP53 is considered to have important roles in the prevention and suppression of abnormal cell proliferation through multiple mechanisms, including cell cycle arrest, apoptosis, and DNA repair [52,53,54]. The Lung Adjuvant Cisplatin Evaluation Biomarker (LACE-Bio) pooled analysis demonstrated that TP53 mutations were marginally predictive of OS benefits from adjuvant platinum-based chemotherapy [54]. However, a study including 197 NSCLC patients enrolled in a randomized trial of postoperative radiation therapy and chemotherapy showed that TP53 mutations and increased expression of P53 protein were not significant prognostic factors in resected stage II–IIIA NSCLC [55]. A previous study suggested that HOXA9 promoter methylation was a prognostic factor that, in combination with mRNA and miRNA-based biomarkers, could identify patients with stage I adenocarcinoma at high risk of recurrence [11]. However, its usefulness is unclear for patients with completely resected NSCLC who have received adjuvant chemotherapy.

ICIs targeting PD-1/PD-L1 have improved the survival of patients with many types of cancer. PD-L1 is currently the most commonly used biomarker for selecting patients who would receive clinical benefits from anti-PD-1/PD-L1 antibodies [56]. Several studies have examined the relationship between PD-L1 expression and the benefit of postoperative adjuvant chemotherapy. A recent study suggested that PD-L1 expression in tumor cells is associated with improved survival with adjuvant chemotherapy (HR, 3.02; 95%CI, 1.69–5.40) [57]. In contrast, the LACE-Bio study showed that neither tumor nor immune cell PD-L1 expression is predictive of clinical benefits from adjuvant chemotherapy [58]. Therefore, it is controversial whether PD-L1 expression in tumor cells is a biomarker for adjuvant chemotherapy in patients with completely resected NSCLC.

Recent studies have attempted a method of verifying cancer characteristics and prognosis by detecting tumor cells and tumor-derived DNA in the blood. Detection and quantification of circulating tumor DNA (ctDNA) by personalized mutation detection panels or cancer personalized profiling by deep sequencing (CAPP-Seq) may help to detect MRD that cannot be detected by imaging [59,60,61]. A recent study showed that postoperative ctDNA positivity is significantly associated with shorter recurrence-free survival. In patients with completely resected stage II–III NSCLC, patients with postoperative ctDNA positivity received benefits from adjuvant chemotherapy, whereas those with postoperative ctDNA negativity had a low risk of relapse without adjuvant chemotherapy [62]. However, its usefulness may be limited in patients with early-stage NSCLC because ctDNA levels in the blood are reported to be associated with tumor size and stage. A study that included 640 cases of various types of cancer showed that ctDNA levels were 100 times higher in stage IV than in stage I disease [63]. The detection sensitivity of ctDNA in early-stage NSCLC is considered to be an issue that requires addressing. CAPP-Seq is a comprehensive mutation analysis method for measurement of ctDNA that is considered to have high sensitivity and may help to detect recurrence of lung cancer in postoperative patients with NSCLC in the future [64]. However, it remains unclear whether measurement of ctDNA using CAPP-Seq is useful as a biomarker for the efficacy of adjuvant chemotherapy in patients with NSCLC. Further studies are needed for the practical use of ctDNA measurement in early-stage NSCLC.

## 4. ACTN4 as a Biomarker for Evaluation of Metastatic Ability

Even if the primary tumor is completely removed grossly, microscopic metastases may remain that cannot be detected by diagnostic imaging. Therefore, indicators of tumor malignancy and high metastatic ability may be useful candidate biomarkers in assessing patient suitability for postoperative adjuvant chemotherapy. Cancer metastasis advances in a multiple-step process. Cancer cells break through the basement membrane, invade the extracellular matrix, and intravasate through the endothelium into the vascular and lymphatic systems to finally establish distant metastatic sites [65,66,67,68]. The dynamic assembly of the actin cytoskeleton plays an important role in the formation processes of cancer metastases. ACTN4 was isolated as a novel isoform of alpha-actinin, which is the actin-binding protein [26,28]. Alpha-actinin has several isotypes in humans, and ACTN4 is classified as the non-muscle type of alpha-actinin [26,28]. Non-muscle types of alpha-actinin, including ACTN4, are related to cell adhesion and cell migration. ACTN-4 has been reported to show increased protein expression in several types of cancers, such as colorectal cancer, pancreatic cancer, ovarian cancer, oral squamous cell carcinoma [29,69,70,71,72], salivary gland carcinoma [73], and lung cancer [28,74,75,76]. Cancer cells at the invasive front of the primal site have high migration and metastatic ability. These cells lose their epithelial cell characteristics, resulting in epithelial–mesenchymal transition [77,78]. Cancer cells at the invasive front show increased expression of ACTN4 protein and EMT-like changes in colorectal cancer tissues [71]. In an in vitro study using cell lines of lung adenocarcinoma, a decrease of ACTN4 expression by siRNA reduced metastatic ability [32]. A previous study showed that siRNA suppression of ACTN-4 protein expression diminished cell protrusion associated with cancer invasion in colon cancer cells [27]. In pancreatic cancer cells, siRNA knockdown of ACTN4 reduced the invasive potential of cancer cells [27]. These results show that ACTN4 is associated with the migration and invasion of cancer cells and plays an important role in the metastasis of cancer. Moreover, ACTN4 directly regulates cell motility by remodeling the actin cytoskeleton [26]. These preclinical studies suggest that ACTN4 may indicate the metastatic ability and malignancy of cancer. Figure 1 shows the possible roles of ACTN4 in cancer metastasis and invasion. ACTN4 mediates the cytoskeleton to sites of cell adhesion and is modulated to enable cell migration [26,79]. ACTN4 has also been reported to induce epithelial–mesenchymal transition (EMT) through upregulation of Snail, which is a transcriptional repressor of E-cadherin expression and one of the main inducers of EMT [80]. Moreover, Snail upregulated by ACTN4 induces cell migration and cancer invasion via Snail-mediated matrix metalloproteinase-9 expression. ACTN4 is involved in the stabilization of β-catenin. The accumulation of β-catenin induced by ACTN4 upregulates cyclin D1 and c-myc, leading to tumorigenesis [81]. Nuclear factor-kappa B (NF-κB) is a transcription factor that regulates cell proliferation, cell differentiation, cellular immunity, and apoptosis [82]. Actinin-4 is related to the transcriptional activity of NF-κB and the NFκB pathway promotes tumor-cell proliferation and survival [83]. NF-κB also plays an important role in both the induction and maintenance of EMT [84]. Further studies are needed to identify the molecular mechanisms of ACTN4 in cancer metastasis and invasion in more detail.

Recent studies using clinical specimens have also examined the usefulness of evaluating ACTN4 in NSCLC. Among these, Miura et al. reported that increased expression of ACTN4 mRNA may be a biomarker for adjuvant chemotherapy in patients with stage IB–II NSCLC. They reported that in a subgroup of patients with increased expression of ACTN4 mRNA, the OS of patients treated with adjuvant cisplatin plus vinorelbine was significantly longer than that of patients who underwent observation without adjuvant chemotherapy (HR, 0.273; 95%CI, 0.079–0.952) [32]. In another study, ACTN4 protein expression was reported to be a promising biomarker in patients with completely resected stage II–IIIA lung adenocarcinoma. In an ACTN4 immunohistochemistry (IHC)-positive subgroup, the OS of patients with adjuvant platinum-based chemotherapy was significantly longer than that of patients without adjuvant platinum-based chemotherapy (HR, 0.307; 95%CI, 0.107–0.882). The five-year relapse-free survival (RFS) rate was 56.5% in patients with adjuvant platinum-based chemotherapy and 33.5% in those without adjuvant platinum-based chemotherapy. In the ACTN4 IHC-negative subgroup, however, there was no significant difference between patients with and without adjuvant platinum-based chemotherapy [33].

Moreover, recent studies have suggested the usefulness of ACTN4 for the evaluation of patients with early-stage lung adenocarcinoma. Even in patients with stage I lung adenocarcinoma, some have recurrence and poor prognosis after curative surgery. *ACTN4* gene amplification is determined by fluorescence in situ hybridization (FISH) in cancer tissues and has been shown to be a prognostic factor in several cancers. Noro et al. reported that *ACTN4* gene amplification was a promising biomarker for predicting the prognosis of chemo-naive patients with stage I adenocarcinoma of the lung, with 5-year DFS and OS rates of patients with *ACTN4* gene amplification of 37% and 64%, respectively. In contrast, the 5-year DFS and OS rates of patients without *ACTN4* gene amplification were 86% and 92%, respectively [85]. In addition to its potential as a prognostic factor, *ACTN4* gene amplification may also be a predictive biomarker for adjuvant UFT therapy in patients with completely resected stage I lung adenocarcinoma. In a retrospective study that included a total of 1136 patients with stage I adenocarcinoma, a subgroup analysis in patients aged ≥ 65 years showed that RFS was significantly longer in the adjuvant UFT therapy group than in the observational group in the *ACTN4* gene amplification positive cohort (HR, 0.084; 95%CI, 0.009–0.806) (Figure 2A) [34]. Among patients who did not receive adjuvant UFT therapy, those with *ACTN4* gene amplification negative had a longer RFS than those with *ACTN4* gene amplification positive (HR, 0.475; 95%CI, 0.239–0.946) (Figure 2B). In contrast, there was no difference in RFS between the adjuvant UFT therapy group and the observational group among *ACTN4* gene amplification negative patients (HR, 0.923; 95%CI, 0.566–1.506) (Figure 2C).

Evaluation of ACTN4 may be beneficial in patients with lung adenocarcinoma as well as those with lung squamous cell carcinoma. The mRNA expression of ACTN4 evaluated by quantitative real-time PCR was a factor significantly associated with cancer-specific mortality in patients with stage I–II lung squamous cell carcinoma (HR, 2.68; 95%CI, 1.21–5.92) [86].

Table 1 summarizes the findings of previous studies that have examined the usefulness of ACTN4 as a predictive or prognostic biomarker in patients with completely resected carcinoma of the lung. These findings suggest that ACTN4 is a promising candidate biomarker for decision-making in postoperative adjuvant chemotherapy in stage I as well as stage II–III patients with NSCLC.

The curves in Figure 3 were created using a Kaplan–Meier plotter, which can assess the relationship between gene expression and survival in a variety of cancers, including lung cancer (https://kmplot.com/analysis/ (accessed on 24 May 2022) [88]. The sources are the Gene Expression Omnibus (GEO), European Genome-phenome Archive (EGA), and The Cancer Genome Atlas (TCGA) databases. Figure 3A,B shows that among lung adenocarcinoma patients, the OS of patients with high ACTN4 is significantly shorter than that of those with low ACTN4. Figure 3C,D shows that among lung squamous cell carcinoma patients, the OS of patients with high ACTN4 is significantly shorter than in those with low ACTN4. These results indicate that ACTN4 is a useful prognostic marker in patients with non-small cell lung cancer regardless of histology.

## 5. Summary of the Advantages and Limitations of Perioperative Biomarkers

Table 2 summarizes the advantages and limitations of candidate biomarkers for the efficacy of adjuvant chemotherapy in lung cancer patients. ERCC1 and TP53 may indicate sensitivity to chemotherapeutic agents including cisplatin rather than MRD [24,52]. Likewise, PD-L1 expression is a predictor of ICI efficacy but not MRD [56]. Therefore, ERCC1, TP53, and PD-L1 may be difficult to use when deciding whether to perform adjuvant chemotherapy, although those biomarkers can be useful for selecting chemotherapeutic agents for perioperative treatment. In contrast, ACTN4 can be useful for decision-making for adjuvant chemotherapy because it indicates tumor metastatic potential and cancer invasiveness rather than sensitivity to specific chemotherapeutic agents [26,28]. CtDNA testing may also be a useful method for predicting postoperative MRD. However, medical cost is one of the challenges regarding its clinical application, as ctDNA testing may increase out-of-pocket expenses for patients [61]. ACTN4 can be evaluated by relatively simple and inexpensive methods such as real-time PCR, IHC, and FISH [32,33,34]. By combining ACTN4 with factors related to sensitivity to certain drugs, such as ERCC1 and PD-L1, it may be possible to provide more appropriate treatment for perioperative patients with lung cancer. The utility of ACTN4 needs to be verified in clinical trials.

## 6. Conclusions

In this review, we have outlined the circumstances under which adjuvant chemotherapy is beneficial in NSCLC and discussed the biological roles of ACTN4 related to cancer invasion and metastases. Increased expression of ACTN4 protein and *ACTN4* gene amplification may be indicators of cancer invasive ability and metastatic ability in all patients with NSCLC. In patients with completely resected NSCLC, ACTN4 may be a useful biomarker of clinical benefit from adjuvant chemotherapy, which may lead to personalized adjuvant chemotherapy.

## Figures and Tables

**Figure 1 cancers-14-04363-f001:**
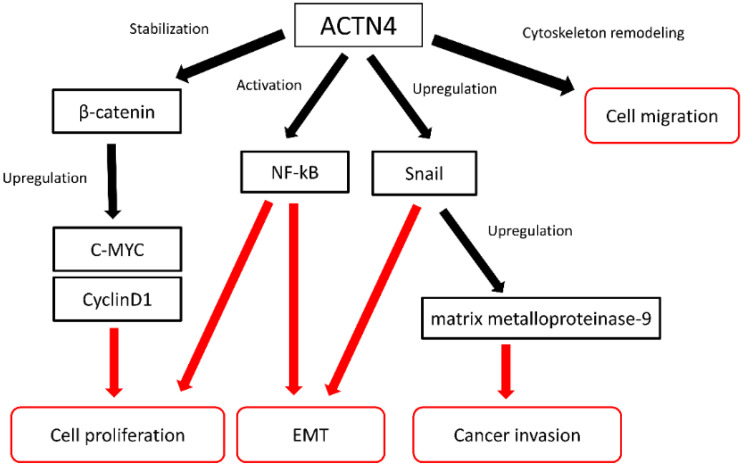
Functions of ACTN4 in cancer metastasis and invasion.

**Figure 2 cancers-14-04363-f002:**
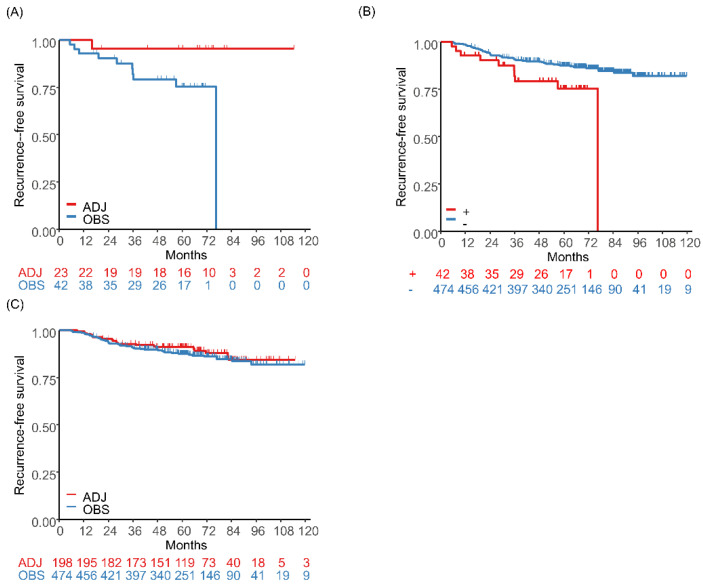
Analyses of patients aged ≥65 years with stage I adenocarcinoma who received adjuvant tegafur/uracil (UFT) therapy or underwent observation. (**A**) comparison of RFS between the adjuvant UFT therapy group and the observational group in patients who were *actinin-4 (ACTN4)* gene amplification positive, (**B**) comparison of RFS between *ACTN4* gene amplification negative and positive patients in the observational group, and (**C**) comparison of RFS between the adjuvant UFT therapy group and the observational group (Noro, R. et al., 2022 [34]).

**Figure 3 cancers-14-04363-f003:**
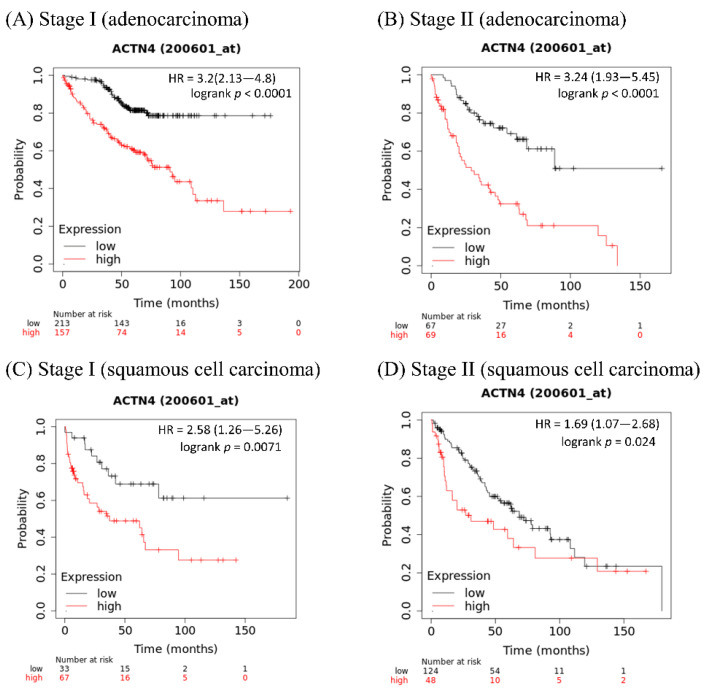
Kaplan–Meier curves showing overall survival in patients with lung cancer (actinin-4 expression high versus low). Overall survival in patients with (**A**) stage I lung adenocarcinoma, (**B**) stage II lung adenocarcinoma, (**C**) stage I lung squamous cell carcinoma, and (**D**) stage II lung squamous cell carcinoma.

**Table 1 cancers-14-04363-t001:** Previous studies of ACTN4 in patients with early-stage non-small cell lung cancer.

	Histology	Stage	Adjuvant Chemotherapy	Evaluation Methods
Miura et al. (2016) [32]	NSCLC	IB-II	CDDP + VNR	mRNA expression
Shiraishi et al. (2017) [33]	Ad	II-IIIA	CDDP + VNR	Protein expression
Noro et al. (2021) [34]	Ad	IA/IB	UFT	Gene amplification
Miyanaga et al. (2013) [75]	HGNT	resected	Not specified	cDNA sequencing
Noro et al. (2013) [85]	Ad	IA-IB	Not specified	Gene amplification
Noro et al. (2017) [86]	Sq	I-II	Not specified	Gene expression
Yamagata et al. (2003) [87]	NSCLC	resected	Not specified	cDNA microarrays

Footnotes: Ad, adenocarcinoma; CDDP, cisplatin; cDNA, complementary DNA; HGNT, high-grade neuroendocrine tumor; NSCLC, non-small cell lung cancer; mRNA, messenger RNA; Sq, squamous cell carcinoma; UFT, tegafur/uracil; VNR, vinorelbine.

**Table 2 cancers-14-04363-t002:** Advantages and limitations of perioperative biomarker candidates.

Biomarker	Function	Advantage	Limitation
Gene expression signature	Gene combinations for poor prognosis and poor chemotherapeutic response	More accurate prognostication of a signature from multiple genes compared with individual genes alone	Statistical validation and reproducibility of the signatures/Not a predictor for MRD
ERCC1	Removal of DNA intrastrand crosslinks by nucleotide excision repair	Predictor for the efficacy of cisplatin	Negative results in randomized phase III clinical trials/Not a predictor for MRD
TP53	Prevention and suppression of abnormal cell proliferation through mechanisms including cell cycle arrest, apoptosis, and DNA repair	One of the most frequently mutated genes in lung cancer regardless of histologic type	Not a predictor for MRD
PD-L1	Binding to its receptor PD-1 expressed by T cells and other immune cells to regulate immune responses	Predictor for the efficacy of anti-PD-1/PD-L1 antibody	Not a predictor for MRD
ctDNA	Tumor-derived DNA released in the blood	Possibility of MRD detection	Cost/Not a predictor for the efficacy of the specific chemotherapeutic agents
ACTN4	Involvement in cancer invasion and metastatic potential	Evaluating tumor metastatic potential and cancer invasiveness	Not a predictor for the efficacy of specific chemotherapeutic agents

Footnotes: ACTN4, actinin-4; ctDNA, circulating tumor DNA; ERCC1, excision repair cross-complementation group 1; MRD, minimal residual disease; PD-1, programmed cell death 1; PD-L1, programmed death-ligand 1; TP53, tumor protein p53 gene.

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
