# Peer review of "Benefits from Adjuvant Chemotherapy in Patients with Resected Non-Small Cell Lung Cancer: Possibility of Stratification by Gene Amplification of *ACTN4* According to Evaluation of Metastatic Ability"

_cancers, 2022, doi:10.3390/cancers14184363_

Round 1

Reviewer 1 Report

Review:

Benefits from adjuvant chemotherapy in patients with resected 2 non-small cell lung cancer: Possibility of stratification by gene amplification of ACTN4 according to evaluation of metastatic ability

In the present review, the authors have highlighted the conditions in which adjuvant chemotherapy is beneficial in NSCLC and taking the opportunity to emphasize the biological potential of ACTN4 gene and protein in cancer invasion and metastases. They showed that overexpression of ACTN4 protein and ACTN4 gene amplification emerge as promise indicators of malignant cells invasive and metastatic capacity in patients with adenocarcinoma and squamous cell carcinoma histotypes of NSCLC. They also emphasized that patients with completely resected NSCLC, ACTN4 may be a useful biomarker of clinical benefit from adjuvant chemotherapy, which may lead to personalized adjuvant chemotherapy.

The review is largely based on the authors' experience and relevant literature. The text is clear and concise using proper English language and grammar. The subject is relevant not only for oncologists but also for clinicians and pathologists. Therefore, I endorse the publication.

Author Response

Comments from reviewer 1

The review is largely based on the authors' experience and relevant literature. The text is clear and concise using proper English language and grammar. The subject is relevant not only for oncologists but also for clinicians and pathologists. Therefore, I endorse the publication.

Answers

Thank you very much for these encouraging comments.

Reviewer 2 Report

Thank you very much for the opportunity for reviewing the manuscript regarding genomic biomarker ACTN4 for metastatic ability for treatment of NSCLC.

I enjoyed to review this manuscript and it is very well written review manuscript.  I just have one comment.

In line 24, it is written as " Previously reported biomarkers of adjuvant chemotherapy for 24 NSCLC are discussed along with their limitations".

If a paragraph is added to the manuscript to support this sentence, it would be nice addition to this review paper.

Author Response

I added the section of Summary of the advantages and limitations of perioperative biomarkers, as below. Thank you very much.

  1. Summary of the advantages and limitations of perioperative biomarkers

Table 2 summarizes the advantages and limitations of candidate biomarkers for the efficacy of adjuvant chemotherapy in lung cancer patients. ERCC1 and TP53 may indicate sensitivity to chemotherapeutic agents including cisplatin rather than MRD (24, 52). Likewise, PD-L1 expression is a predictor of ICI efficacy, and not MRD (56). Therefore, ERCC1, TP53, and PD-L1 may be difficult to use when deciding whether to perform adjuvant chemotherapy, although those biomarkers can be useful for selecting chemotherapeutic agents for perioperative treatment. In contrast, ACTN4 can be useful for decision-making for adjuvant chemotherapy because it indicates tumor metastatic potential and cancer invasiveness rather than sensitivity to specific chemotherapeutic agents (26, 28). CtDNA testing may also be a useful method for predicting postoperative MRD. However, medical cost is one of the challenges regarding its clinical application, as ctDNA testing may increase out-of-pocket expenses for patients (61). ACTN4 can be evaluated by relatively simple and inexpensive methods such as real-time PCR and IHC, and FISH (32-34). By combining ACTN4 with factors related to sensitivity to certain drugs, such as ERCC1 and PD-L1, it may be possible to provide more appropriate treatment for perioperative patients with lung cancer. The utility of ACTN4 needs to be verified in clinical trials.

Reviewer 3 Report

The below points must be considered before this review can be accepted.

Specific points for the authors:

a) Expand statement in lines 51-54 with citation(s), better link others' work, but no suggestions like "should be----".

b) Lines 87, 91, 96, 97, 147, 149, 158, 159,  citations are missing.

c) I suggest using elevated or increased expression rather than over-expression. This is because term over-expression is used for ectopic expression of the desired gene.

d) Grammatical errors: There are lots of spelling mistakes. Therefore, I recommend using professional proofreading services.

Author Response

  1. Expand statement in lines 51-54 with citation(s), better link others' work, but no suggestions like "should be----".

Thank you very much for this comment.

We have added some relevant citations and expanded the discussion in these sentences. In addition, we have changed the wording according your suggestions.

Moreover, the IMpower 010 trial demonstrated that atezolizumab after adjuvant platinum-based chemotherapy improved DFS in postoperative patients who were selected by programmed death-ligand 1 [PD-L1] expression [6]. The ADAURA trial demonstrated that osimertinib improved DFS in epidermal growth factor receptor (EGFR) mutation-positive stage IB–IIIA disease following complete resection [7]. However, a pooled analysis of several clinical trials showed that some patients without adjuvant chemotherapy achieved five-year DFS and that the rate of overall grade 3 to 4 toxicity was 66% in patients with adjuvant platinum-based chemotherapy [5]. Therefore, some patients were cured of lung cancer by chest surgery alone and adjuvant chemotherapy may be overtreatment for such patients [5]. Economic analysis of a randomized trial of adjuvant vinorelbine plus cisplatin suggested high cost-effectiveness of this combined treatment compared with other standard health care interventions [8]. However, a pilot study revealed that patients undergoing cancer treatment may change their treatment to defray out-of-pocket costs be-cause of financial burden [9]. In particular, molecularly targeted therapies and immune checkpoint inhibitors (ICIs) may increase economic toxicity for cancer patients due to the high prices of these drugs [10]. From the perspective of reducing adverse events and saving on the medical costs of adjuvant chemotherapy, biomarkers that can predict the likelihood of recurrence after surgery would be of great interest.

  1. Lines 87, 91, 96, 97, 147, 149, 158, 159, citations are missing.

We have added the appropriate citations, accordingly.

  1. I suggest using elevated or increased expression rather than over-expression. This is because term over-expression is used for ectopic expression of the desired gene.

We have changed the wording throughout the manuscript, as suggested.

  1. d) Grammatical errors: There are lots of spelling mistakes. Therefore, I recommend using professional proofreading services.

The manuscript has now been checked by a professional native-English-speaking editor.

Reviewer 4 Report

We read with great interest the Tozuka et al where the authors

 review the current status of adjuvant chemotherapy in non-small cell lung cancer (NSCLC) and describe promising perioperative therapies, including molecularly targeted therapies  and immune checkpoint inhibitors. The authors discuss the potential utility of biomarkers in patient stratification for effective adjuvant chemotherapy for NSCLC. And for this purpose, they will evaluate the role of actinin-4 (ACTN4) as a possible biomarker for NSCLC.

There are minor comments:

the review is highly timely and is of high interest to researchers and clinicians in the field of cancer. The review article is very focused with Well written sections having good flow. While the authors describe the recurrence of NSCLC where some patients with completely resected NSCLC may have minimal residual disease (MRD) and the use of the biomarker “excision repair cross-complementation group 1 (ERCC1)”, as candidate biomarkers for adjuvant 60 chemotherapy in patients with NSCLC; it was not clear what makes ACTN4 the “One” biomarker to be specific for NSCLC compared to the other candidates discussed in the paper. This section needs to be added to highlight the prospective value of ACTN4 compared to other biomarkers. Also, it would be helpful to include a section about the limitation of ACTN4 as a biomarker  

I would also suggest that the authors present a table for the different potential markers and present their utility; this will give a more comprehensive depiction of different biomarkers of the NSCLC.

Author Response

Reviewer 3

Comments from reviewer 3

This section needs to be added to highlight the prospective value of ACTN4 compared to other biomarkers. Also, it would be helpful to include a section about the limitation of ACTN4 as a biomarker 

I would also suggest that the authors present a table for the different potential markers and present their utility; this will give a more comprehensive depiction of different biomarkers of the NSCLC.

Answers

Thank you very much for these helpful suggestions. We have added a section titled “Summary of the advantages and limitations of perioperative biomarkers”, and discussed the advantages and limitations of each biomarker in the text and in Table 2.

  1. Summary of the advantages and limitations of perioperative biomarkers

Table 2 summarizes the advantages and limitations of candidate biomarkers for the efficacy of adjuvant chemotherapy in lung cancer patients. ERCC1 and TP53 may indicate sensitivity to chemotherapeutic agents including cisplatin rather than MRD (24, 52). Likewise, PD-L1 expression is a predictor of ICI efficacy, and not MRD (56). Therefore, ERCC1, TP53, and PD-L1 may be difficult to use when deciding whether to perform adjuvant chemotherapy, although those biomarkers can be useful for selecting chemotherapeutic agents for perioperative treatment. In contrast, ACTN4 can be useful for decision-making for adjuvant chemotherapy because it indicates tumor metastatic potential and cancer invasiveness rather than sensitivity to specific chemotherapeutic agents (26, 28). CtDNA testing may also be a useful method for predicting postoperative MRD. However, medical cost is one of the challenges regarding its clinical application, as ctDNA testing may increase out-of-pocket expenses for patients (61). ACTN4 can be evaluated by relatively simple and inexpensive methods such as real-time PCR and IHC, and FISH (32-34). By combining ACTN4 with factors related to sensitivity to certain drugs, such as ERCC1 and PD-L1, it may be possible to provide more appropriate treatment for perioperative patients with lung cancer. The utility of ACTN4 needs to be verified in clinical trials.

Table 2. Advantages and limitations of perioperative biomarker candidates.

Biomarker

Function

Advantage

Limitation

Gene expression signature

Gene combinations for poor prognosis and poor chemotherapeutic response

More accurate prognostication of a signature from multiple genes compared with individual genes alone

Statistical validation and reproducibility of the signatures/Not a predictor for MRD

ERCC1

Removal of DNA intrastrand crosslinks by nucleotide excision repair

Predictor for the efficacy of cisplatin

Negative results in randomized phase III clinical trials/Not a predictor for MRD

TP53

Prevention and suppression of abnormal cell proliferation through mechanisms including cell cycle arrest, apoptosis, and DNA repair

One of the most frequently mutated genes in lung cancer regardless of histologic type

Not a predictor for MRD

PD-L1

Binding to its receptor PD-1 expressed by T cells and other immune cells to regulate immune responses

Predictor for the efficacy of anti-PD-1/PD-L1 antibody

Not a predictor for MRD

ctDNA

Tumor-derived DNA released in the blood

Possibility of MRD detection

Cost/Not a predictor for the efficacy of specific chemotherapeutic agents

ACTN4

Involvement in cancer invasion and metastatic potential

Evaluating tumor metastatic potential and cancer invasiveness

Not a predictor for the efficacy of specific chemotherapeutic agents

Footnotes: ACTN4, actinin-4; ctDNA, circulating tumor DNA; ERCC1, excision repair cross-complementation group 1; MRD, minimal residual disease; PD-1, programmed cell death 1; PD-L1, programmed death-ligand 1; TP53, tumor protein p53 gene